# The Effects of Binding Moral Foundations on Prejudiced Attitudes toward Migrants: The Mediation Role of Perceived Realistic and Symbolic Threats

Fleur Bianco and Ankica Kosic *

Faculty of Medicine and Psychology, Sapienza University of Rome, Via dei Marsi 78, 00185 Rome, Italy
* Correspondence: anna.kosic@uniroma1.it

**Abstract:** (1) Background: This study explores how threat perception mediates the relationship between binding moral foundations and prejudice toward migrants. We hypothesized that the relationship between binding moral foundations and prejudice against migrants, which is already established in the literature, is mediated through realistic and symbolic threat perception. (2) Methods: Two separate samples were gathered, in Malta ($N = 191$) and Italy ($N = 189$). The participants responded to an anonymous questionnaire containing several scales: the Moral Foundation Questionnaire, perceived threat from migrants, prejudice toward migrants, and social distance from several macro-categories of migrants. (3) Results: We confirmed a significant relationship between binding moral foundations and explicit prejudice toward migrants, and also found that this relationship was mediated by perceived realistic and symbolic threats in both countries. However, when the indices of social distance were considered as criterion variables, the direct relationship between binding moral foundations and social distance was not confirmed for all the migrant macro-groups. In addition, in some migrant groups, we found that this relationship was mediated by perceived realistic and symbolic threats. (4) Conclusions: This study indicates that the perception of realistic threats may have a significant role in determining the effect of binding moral foundations; this may have theoretical and practical implications.

**Keywords:** binding moral foundations; perceived realistic and symbolic threats; prejudice toward migrants; social distance

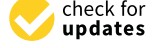



## 1. Introduction

With an increase in migrant populations across different regions of the world, investigating the factors that explain prejudice and racist incidents is essential. Previous studies found that loyalty, authority and purity (binding moral foundations) toward the ingroup contribute to increased levels of prejudice against outgroup members (e.g., Baldner and Pierro 2019; Kugler et al. 2014) and have shown that outgroups are perceived as threatening. These studies have not, however, investigated the role of perceived threat in the relationship between the binding moral foundations and prejudice toward migrants. Moreover, previous studies have focused on attitudes toward immigrants in general, considering them as a homogeneous group, without analyzing whether there is heterogeneity in attitudes toward different migrant groups. To address this gap, we sought to find results supporting the assumption that the perception of realistic and symbolic threats mediates the relationship between binding morality and prejudice toward migrants in two countries: Malta and Italy. More specifically, the study aims to investigate whether the binding moral foundations predict prejudice toward various migrant groups to the same extent or if that relationship is mediated to a different extent by the perceived threat that may be posed by these groups.

## 1.1. Moral Foundations

Moral Foundations Theory (MFT) attempts to integrate the motives and values of moral intuitions and judgments (Graham et al. 2011; Haidt and Graham 2007). According to MFT, moral intuitions are determined by individual moral foundations that are rooted in our evolutionary process, and shaped by the social and cultural environment people live in. It delineates five different morality domains: Care–Harm, Fairness–Cheating, known as the individualizing foundation, and Loyalty–Betrayal, Authority–Subversion, and Sanctity–Degradation, known as the binding foundation (Graham et al. 2009, 2011).

Protecting the community from individual selfishness is the main evolutionary process of both these foundations. However, these foundations employ different ways to achieve this. While the individualizing foundations seek to protect from selfishness by disapproving of harm and cheating among individuals, the binding moral foundations protect communities by fostering respect for authority, loyalty toward the in-group, and purity (Kugler et al. 2014).

Individuals' preferred moral foundations can predict whether people favor or dislike members of outgroups (Graham et al. 2011). Therefore, it follows that the different functions of the foundations have different implications for intergroup behaviors and attitudes. The host group may fear losing ingroup customs, language, and traditions due to interaction with migrants. Since moral foundations are an antecedent of threat and have been portrayed as fundamental and stable aspects of morality that are difficult to alter (Haidt and Graham 2007), we reason that these may influence individuals' worldviews and behavior. In effect, several studies have found that the binding moral foundations are associated with prejudice, social distancing, and avoidance of outgroups (e.g., Baldner and Pierro 2019; Bianco and Kosic 2022; Bianco et al. 2021). These studies focused on attitudes toward immigrants in general, considering them as a homogeneous group. However, public attitudes may differ depending on the ethnicity of immigrants. Several studies in Europe, the United States, and Canada showed that immigrants of Northern European origin were ranked at the top of the hierarchy, followed by Southern and Eastern Europeans, while Asian and African immigrants were put at the bottom of the hierarchy (e.g., Brader et al. 2008; Gorodzeisky and Semyonov 2019; Tartakovsky and Walsh 2020; Turper et al. 2015; for a review, see Hagendoorn 1995). Similarly, Ford (2011) found that in Great Britain immigrants from Western Europe and Australia encountered less public hostility than immigrants from Africa and South Asia. Bridges and Mateut (2014) showed that Europeans were more inclined to reject immigrants from racial minority groups. Various studies have suggested that different minority groups can elicit different perceptions of threat and prejudice (e.g., Bessudnov 2016; Brunarska and Soral 2022; Hellwig and Sinno 2017). Following this literature, our study investigates the relationship between the binding moral foundations and prejudice toward migrants in Italy and Malta who come from different regions of the world. In addition, the study explores whether this relationship may be mediated by the perceived threat posed by migrants.

## 1.2. Perceived Realistic and Symbolic Threats

According to the generalized prejudice approach (Bergh et al. 2016), sometimes known as group-focused enmity in the literature (Zick et al. 2008), attitudes toward various outgroups have a common basis. In other words, these theoretical perspectives suggest that some people demonstrate a generalized prejudice against several target groups. Conversely, the differentiated threat approach suggests that prejudice depends on the source of the threat posed by a certain outgroup (Meuleman et al. 2018; see also Cottrell and Neuberg 2005).

The literature usually divides the sources of outgroup threat into two main categories—realistic and symbolic (according to the Revised Integrated Threat Theory; Stephan et al. 2009). These are sometimes also considered as material versus cultural threats (Bloom et al. 2015). Realistic threat implies the perception of competition between groups for scarce resources, such as education, jobs, social services, etc. (Vallejo-Martín et al. 2020). Symbolic threat refers

to the conviction that outgroup members challenge the ingroup's values, beliefs, culture, or way of life.

In the literature, perceived realistic and symbolic threats have been associated with increased hostility, dislike, disregard, rejection, hatred, and superiority toward migrants (e.g., Stephan et al. 1999). Stephan and Stephan (2016) argue that intergroup threat serves as a mediator between its cause and consequence. They suggest that this variable shows up as both cause and effect and is therefore reciprocal: negative attitudes toward a particular outgroup and its members may cause the ingroup to perceive a threat from outgroups, but similarly, perceptions of ingroup threat can also heighten negative attitudes toward some outgroups.

*1.3. Aims*

Accordingly, this study has the following objectives: (1) to explore attitudes toward migrants (in terms of explicit generalized prejudice and social distance from several groups of migrants from different regional provenience); (2) to investigate whether the binding moral foundations have the same predictive validity of prejudice toward different groups of migrants; and (3) to test whether the relationship between the binding moral foundations and prejudice is mediated by perceived realistic and symbolic threats for different groups of migrants. We hypothesize that the binding moral foundations will predict the perception of realistic and symbolic threats but with different strengths for different categories of migrants. In line with the differentiated threat approach, migrants coming from different places or regions "challenge diverse realistic and symbolic interests" (Meuleman et al. 2018), and therefore they are not perceived uniformly by the receiving society. Thus, we may expect that the relationship between the binding moral foundations and prejudice toward different migrant groups (measured as social distance) could be different for different categories of migrants. In addition, people with high binding moral foundations are found to be more sensitive to social dangers (e.g., Van Leeuwen and Park 2009). Thus, the relationship between binding foundations and perceived threat will be stronger for some groups that are perceived as a major threat to ingroup stability, norms of respect for authority, loyalty, and sense of purity.

*1.4. Mapping the Migration Context in Malta and Italy*

Increasing numbers of migrants have settled in both countries. Malta leads the board with 14.1% of the total population being migrants, while in Italy, 8.5% are migrants. Around 39.9 million foreign citizens reside within the European Union's borders, of whom 22.3 million originate from non-member countries. Non-European Union member migration is higher in Italy (69.9%) than in Malta (39%) (Eurostat 2019).

## 2. Results

*2.1. Correlations and Descriptive Statistics*

Correlations and descriptive statistics are reported in Tables 1 and 2. We can see that in Malta the index of the binding moral foundations is correlated positively and in a similar way to explicit prejudice and social distance for all the groups of migrants (it is slightly higher when considering Hispanic migrants), while in Italy this correlation is not significant for the indexes of social distance for Hispanic migrants and for Western European/American migrants. The index of the binding foundations is correlated significantly with both types of perceived threats (realistic and symbolic) in both countries. The index of the individualizing moral foundations is not significantly correlated with the indexes of either prejudice or social distance in Malta, whereas in Italy it is correlated significantly and negatively with explicit prejudice and social distance from African, Hispanic, and Western European/American migrants.

**Table 1.** Descriptive statistics (mean and standard deviation) and correlations (Malta: *N* = 191).

| Variables | M | SD | 1 | 2 | 3 | 4 | 5 | 6 | 7 | 8 | 9 | 10 |
|---|---|---|---|---|---|---|---|---|---|---|---|---|
| 1. BMF | 4.18 | 0.82 | - | | | | | | | | | |
| 2. PRT | 3.57 | 1.31 | 0.43 ** | - | | | | | | | | |
| 3. PST | 3.57 | 1.61 | 0.42 ** | 0.75 ** | - | | | | | | | |
| 4. PREJ | 2.71 | 1.11 | 0.43 ** | 0.72 ** | 0.65 ** | - | | | | | | |
| 5. ASIA | 2.41 | 1.31 | 0.28 ** | 0.47 ** | 0.44 ** | 0.54 ** | - | | | | | |
| 6. EEU | 2.74 | 1.59 | 0.32 ** | 0.48 ** | 0.46 ** | 0.43 ** | 0.64 ** | - | | | | |
| 7. AFR | 3.15 | 1.59 | 0.31 ** | 0.62 ** | 0.52 ** | 0.55 ** | 0.66 ** | 0.65 ** | - | | | |
| 8. HIS | 2.51 | 1.46 | 0.37 ** | 0.50 ** | 0.48 ** | 0.44 ** | 0.64 ** | 0.62 ** | 0.63 ** | - | | |
| 9. MEA | 3.02 | 1.56 | 0.29 ** | 0.59 ** | 0.52 ** | 0.57 ** | 0.65 ** | 0.60 ** | 0.89 ** | 0.62 ** | - | |
| 10. W/A | 2.05 | 1.29 | 0.31 ** | 0.39 ** | 0.39 ** | 0.36 ** | 0.53 ** | 0.59 ** | 0.48 ** | 0.64 ** | 0.42 ** | - |
| 11. IMF | 4.78 | 0.52 | 0.44 ** | −0.01 | −0.02 | −0.09 | 0.02 | 0.10 | 0.00 | 0.06 | −0.00 | 0.07 |

Note: ** $p < 0.001$. Legend: BMF = binding moral foundations; PRT = perceived realistic threat; PST = perceived symbolic threat; PREJ = prejudice; ASIA = social distance from Asian migrants; EEU = social distance from East European migrants; AFR = social distance from African migrants; HIS = social distance from Hispanic migrants; MEA = social distance from Middle Eastern migrants; W/A = social distance from West European/American migrants; IMF = individualizing moral foundations.

**Table 2.** Descriptive statistics (mean and standard deviation) and correlations (Italy: *N* = 189).

| Variables | M | SD | 1 | 2 | 3 | 4 | 5 | 6 | 7 | 8 | 9 | 10 |
|---|---|---|---|---|---|---|---|---|---|---|---|---|
| 1. BMF | 3.65 | 0.78 | - | | | | | | | | | |
| 2. PRT | 2.68 | 1.27 | 0.42 ** | - | | | | | | | | |
| 3. PST | 3.05 | 1.07 | 0.27 ** | 0.66 ** | - | | | | | | | |
| 4. PREJ | 2.13 | 1.10 | 0.33 ** | 0.74 ** | 0.63 ** | - | | | | | | |
| 5. ASIA | 2.24 | 1.40 | 0.29 ** | 0.44 ** | 0.33 ** | 0.40 ** | - | | | | | |
| 6. EEU | 2.57 | 1.56 | 0.30 ** | 0.44 ** | 0.39 ** | 0.50 ** | 0.73 ** | - | | | | |
| 7. AFR | 2.43 | 1.49 | 0.28 ** | 0.53 ** | 0.51 ** | 0.63 ** | 0.76 ** | 0.81 ** | - | | | |
| 8. HIS | 2.05 | 1.34 | 0.14 | 0.37 ** | 0.32 ** | 0.46 ** | 0.72 ** | 0.68 ** | 0.77 ** | - | | |
| 9. MEA | 2.39 | 1.47 | 0.31 ** | 0.56 ** | 0.47 ** | 0.61 ** | 0.77 ** | 0.77 ** | 0.85 ** | 0.70 ** | - | |
| 10. W/A | 1.73 | 1.21 | 0.11 | 0.24 ** | 0.22 ** | 0.30 ** | 0.56 ** | 0.47 ** | 0.52 ** | 0.69 ** | 0.54 ** | - |
| 11. IMF | 4.38 | 0.68 | 0.38 ** | −0.05 | −0.21 ** | −0.26 ** | −0.08 | −0.15 | −0.20 ** | −0.26 ** | −0.14 | −0.22 ** |

Note: ** $p < 0.001$. Legend: BMF = binding moral foundations; PRT = perceived realistic threat; PST = perceived symbolic threat; PREJ = prejudice; ASIA = social distance from Asian migrants; EEU = social distance from East European migrants; AFR = social distance from African migrants; HIS = social distance from Hispanic migrants; MEA = social distance from Middle Eastern migrants; W/A = social distance from West European/American migrants; IMF = individualizing moral foundations.

When observing the mean values, we can see that in Malta the participants had higher social distance from African and Middle Eastern migrants. In Italy, however, we found higher social distance from East European migrants, Africans, and Middle Eastern migrants. To check the differences between countries, we carried out a one-way ANOVA. Before conducting the ANOVA, we confirmed that the data were normally distributed in both countries. The distributions associated with skewness and kurtosis were between −2 and +2 (George and Mallery 2010). We performed an ANOVA Welch Test of homogeneity of variance using Levene's F test (see Table 3). We found higher values in Malta for most of the variables, especially for binding and individualizing foundations, explicit prejudice, and social distance for African, Asian, Middle Eastern, and Western European/American migrants. There were no significant differences for social distance from East European and Hispanic migrants.

**Table 3.** One-way analyses of variance between the two countries(Malta and Italy *N* = 380).

| | |
|---|---|
| 1. Percevied Realistic Threat | $F(1,378) = 4.17, p = 0.042$ |
| 2. Perceived Symbolic Threat | $F(1,378) = 5.17, p = 0.024$ |
| 3.Binding Moral Foundations | $F(1,378) = 42.28, p = 0.000$ |
| 4. Individualizing Moral Foundations | $F(1,378) = 41.75, p = 0.000$ |
| 5. Explicit Prejudice | $F(1,378) = 151.33, p = 0.000$ |
| 6. Social Distance from Asian migrants | $F(1,378) = 35.40, p = 0.000$ |
| 7. Social distance from East European migrants | $F(1,378) = 0.17, p = 0.682$ |
| 8. Social distance from African migrants | $F(1,378) = 14.09, p = 0.000$ |
| 9. Social distance from Hispanic migrants | $F(1,378) = 0.00, p = 0.966$ |
| 10. Social distance from Middle Eastern migrants | $F(1,378) = 67.16, p = 0.000$ |
| 11. Social distance from West European/American migrants | $F(1,378) = 227.86, p = 0.000$ |

### 2.2. Parallel Mediation Analysis

Here, we test our hypothesized model that the binding moral foundations predict prejudice toward migrants (and social distance from several groups of migrants), and analyze whether this relationship may be mediated by perceived realistic and symbolic threats differently for different groups of migrants. We performed a mediation analysis (PROCESS; Model 4, Hayes 2017) with the 5000 bootstrap method, and with bias correction 95% confidence intervals (*CI's*) for indirect effects (Preacher and Hayes 2008). Binding moral foundation is an independent variable (X), perceived realistic threat (M1), and perceived symbolic threat (M2) are the mediators, and prejudice toward migrants and indexes of social distance from the six groups of migrants are the criterion variables (Y). We conducted this analysis seven times for each sample. All the variables were standardized.

Initially, we checked for possible covariates (e.g., individualizing moral foundations, age, and education level). However, we found that age and education had little or no effect on the mediators or dependent variables in both samples. Consequentially, we ran the model using only the index of the individualizing moral foundations as a covariate.

### 2.3. Explicit Prejudice against Migrants

The total effect of the binding foundations was significant as follows: Malta: $b = 0.60$, $SE = 0.07$, $F(2,188) = 39.91$, $p < 0.001$, 95% *CI* [0.46, 0.73]; and Italy: $b = 0.49$, $SE = 0.07$, $F(2,188) = 35.18$, $p < 0.001$, 95% *CI* [0.36,0.63]. The direct effect was also significant as follows: Malta: $b = 0.23$, $SE = 0.06$, $p < 0.001$, 95% *CI* [0.11, 0.34]; and Italy: $b = 0.14$, $SE = 0.06$, $p < 0.001$, 95% *CI* [0.03,0.25].

In both countries, the binding moral foundations positively and significantly predicted perceived realistic and symbolic threats. Moreover, perceived realistic and symbolic threats positively predicted prejudice against migrants. We found a direct negative effect of the individualizing moral foundations on prejudice toward migrants in both countries (see Figures 1 and 2).

Indirect Effects

A significant, although small, indirect effect of the binding moral foundations was found on explicit prejudice through perceived realistic and symbolic threats in both countries. Perceived realistic threat, however, seemed to be the most relevant predictor (see Table 4). These results confirm a partial mediation. In the Maltese sample, $R^2 = 0.30$, and in the Italian sample, $R^2 = 0.27$, meaning that, respectively, 30% and 27% of the variance of the explicit prejudice against migrants is explained by the hypothesized relationships.

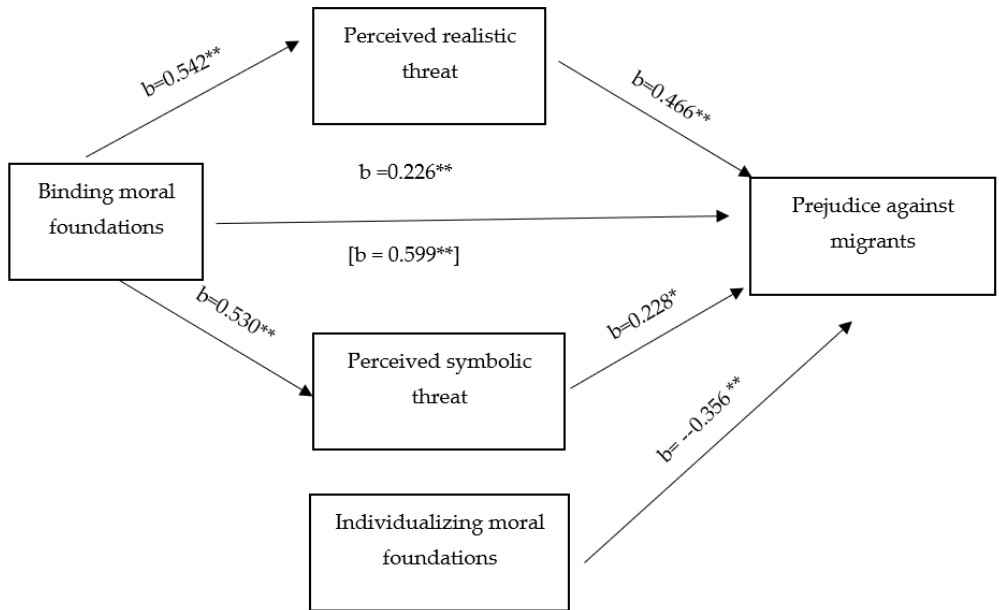

**Figure 1.** Path analysis of hypothesized model, Malta (*N* = 191). Note: ** *p* < 0.001, * *p* < 0.005.

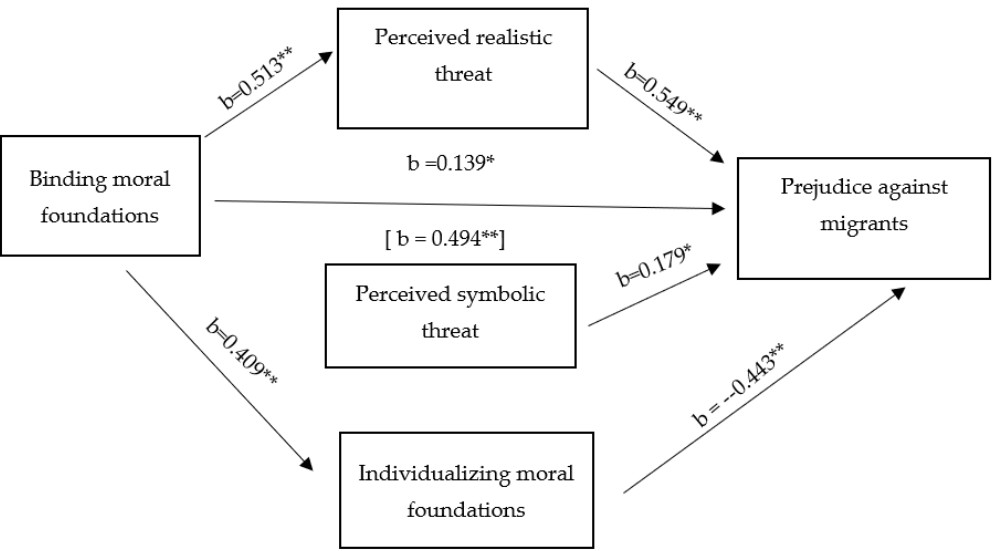

**Figure 2.** Path analysis of hypothesized model, Italy (*N* = 189). Note: ** *p* < 0.001, * *p* < 0.005.

**Table 4.** Standardized indirect effects of binding moral foundations on explicit prejudice.

|  | Through Perceived Realistic Threat Indirect; SE; 95% CI LL;UL | Through Perceived Symbolic Threat Indirect; SE; 95% CI LL;UL |
|---|---|---|
| Malta | 0.25; 0.06 [0.15; 0.37] | 0.12; 0.05 [0.04; 0.22] |
| Italy | 0.28; 0.05 [0.18; 0.39] | 0.07; 0.04 [0.01; 0.15] |

Legend: *CI* = confidence interval; *LL* = lower limit; *UL* = upper limit.

### 2.4. Social Distance from Migrant Groups

We found a significant total effect for all six migrant groups in Malta as follows: Asian: *b* = 0.33; *SE* = 0.08; *p* < 0.001; 95%; *CI* [0.18; 0.49]; East European: *b* = 0.35; *SE* = 0.08; *p* < 0.001; 95%; *CI* [0.19; 0.50]; African: *b* = 0.38; *SE* = 0.07; *p* < 0.001; 95%; *CI* [0.23; 0.53]; Hispanic: *b* = 0.42; *SE* = 0.08; *p* < 0.001; 95%; *CI* [0.27; 0.57]; Middle Eastern: *b* = 0.36; *SE* = 0.08; *p* < 0.001; 95%; *CI* [0.21; 0.51]; and West European/American: *b* = 0.34; *SE* = 0.08; *p* < 0.001; 95%; *CI* [0.19; 0.50].

Similar effects were found in Italy as follows: Asian: $b = 0.37$; $SE = 0.07$; $p < 0.001$; 95%; *CI* [0.23; 0.52], East European: $b = 0.42$; $SE = 0.07$; $p < 0.001$; 95%; *CI* [0.28; 0.56]; African: $b = 0.41$; $SE = 0.07$; $p < 0.001$; 95%; *CI* [0.27; 0.55]; Hispanic: $b = 0.28$; $SE = 0.07$; $p < 0.001$; 95%; *CI* [0.13; 0.42]; Middle Eastern: $b = 0.43$; $SE = 0.07$; $p < 0.001$; 95%; *CI* [0.29; 0.57]; and West European/American: $b = 0.22$; $SE = 0.08$; $p < 0.005$; 95%; *CI* [0.07; 0.37].

A direct effect of the binding moral foundations on social distance(prejudice) was found only for the Hispanic migrants in Malta. In Italy, however, a direct effect was found for the Asian, East European, African, and Middle Eastern migrants (see Table 5).

**Table 5.** The relationship between moral foundations, perceived threats and social distance for several migrant groups.

| | Binding MF | | | Perceived Realistic Threat | | | Perceived Symbolic Threat | | | Individualizing MF | | |
|---|---|---|---|---|---|---|---|---|---|---|---|---|
| | *b* | *SE* | *p* | *b* | *SE* | *p* | *b* | *SE* | *p* | *b* | *SE* | *p* |
| MALTA | | | | | | | | | | | | |
| ASIA | 0.07 | 0.07 | 0.387 | 0.30 | 0.10 | <0.005 | 0.19 | 0.10 | 0.060 | −0.01 | 0.07 | 0.921 |
| EEU | 0.07 | 0.08 | 0.350 | 0.30 | 0.10 | <0.005 | 0.20 | 0.10 | <0.05 | −0.08 | 0.07 | 0.302 |
| AFRICA | 0.04 | 0.07 | 0.557 | 0.50 | 0.09 | <0.001 | 0.13 | 0.09 | 0.156 | −0.01 | 0.07 | 0.889 |
| HISPANIC | 0.17 | 0.08 | <0.05 | 0.27 | 0.10 | <0.01 | 0.20 | 0.10 | <0.05 | −0.01 | 0.07 | 0.927 |
| MEA | 0.03 | 0.08 | 0.716 | 0.44 | 0.09 | <0.001 | 0.18 | 0.09 | 0.052 | −0.01 | 0.07 | 0.908 |
| W/A | 0.15 | 0.09 | 0.091 | 0.18 | 0.10 | 0.074 | 0.19 | 0.10 | 0.069 | 0.01 | 0.08 | 0.934 |
| ITALY | | | | | | | | | | | | |
| ASIA | 0.19 | 0.08 | <0.05 | 0.33 | 0.07 | <0.001 | 0.03 | 0.08 | 0.729 | −0.13 | 0.08 | 0.098 |
| EEU | 0.25 | 0.08 | <0.005 | 0.25 | 0.09 | <0.01 | 0.11 | 0.09 | 0.212 | −0.21 | 0.07 | <0.01 |
| AFRICA | 0.17 | 0.07 | <0.05 | 0.32 | 0.08 | <0.001 | 0.21 | 0.08 | <0.05 | −0.20 | 0.07 | <0.005 |
| HISPANIC | 0.11 | 0.08 | 0.164 | 0.28 | 0.09 | <0.005 | 0.05 | 0.09 | 0.601 | −0.28 | 0.08 | <0.001 |
| MEA | 0.18 | 0.07 | <0.05 | 0.39 | 0.08 | <0.001 | −13 | 0.08 | 0.103 | −0.17 | 0.07 | <0.05 |
| W/A | 0.13 | 0.09 | 0.127 | 0.15 | 0.10 | 0.126 | 0.03 | 0.10 | 0.789 | −0.25 | 0.08 | <0.005 |

Legend: Binding MF = binding moral foundations; individualizing MF = individualizing moral foundations. ASIA = social distance from Asian migrants; EEU = social distance from East European migrants; AFRICA = social distance from African migrants; HISPANIC = social distance from Hispanic migrants; MEA = social distance from Middle Eastern migrants; W/A = social distance from West European/American migrants.

These results in the Maltese sample suggest a partial mediation for the Hispanic migrants and full mediation in the remaining migrant groups, whereas in the Italian sample a partial mediation emerged for the Asian, East European, African, and Middle Eastern migrants, and full mediation for the Hispanic and West European/American migrants.

The effects of the binding moral foundations on perceived realistic and symbolic threats were significant in both samples. While the effect of perceived realistic threat on social distance was significant for all migrant groups except the Western European/American migrant group in both samples, perceived symbolic threat was significant only for the East European and Hispanic migrant groups in the Maltese sample, and for the African and Western European/American migrant groups in the Italian sample.

Besides these results, a direct negative effect of the individualizing moral foundations on all categories of migrants was found in both samples, except for the Western European/American migrants in the Maltese sample (Table 5).

Indirect Effects

In both samples, we found an indirect effect of the binding moral foundations through perceived realistic threat on social distance from all the migrant groups, except from the West European/American migrants.

No indirect effects through perceived symbolic threat were found in the Maltese sample, whilst in the Italian sample an indirect effect was found for the African migrants. These results imply that, when mediated by perceived realistic threat, the relationship between the binding moral foundations and prejudice is significant for the majority of migrant groups considered in this study, but it is not the case when perceived symbolic threat is considered as a mediator (see Table 6).

**Table 6.** Indirect effects of binding moral foundations on social distance from different migrant groups.

| | Through Perceived Realistic Threat Indirect; SE; 95% CI LL; UL | Through Perceived Symbolic Threat Indirect; SE; 95% CI LL; UL |
|---|---|---|
| **MALTA** | | |
| Asia | 0.16; 0.06 [0.14; 0.36] | 0.10; 0.06 [−0.02; 0.22] |
| East Europe | 0.16; 0.06 [0.05; 0.30] | 0.11; 0.06 [−0.02; 0.23] |
| African | 0.27; 0.07 [0.16; 0.42] | 0.07; 0.06 [−0.05; 0.18] |
| Hispanic | 0.15; 0.06 [0.04; 0.33] | 0.11; 0.06 [−0.01; 0.23] |
| Middle East | 0.24; 0.06 [0.12; 0.37] | 0.09; 0.06 [−0.02; 0.21] |
| West European/American | 0.10; 0.06 [−0.00; 0.26] | 0.10; 0.07 [−0.02; 0.23] |
| **ITALY** | | |
| Asia | 0.17; 0.05 [0.07; 0.28] | 0.13; 0.04 [−0.06; 0.10] |
| East Europe | 0.03; 0.05 [0.03; 0.24] | 0.05; 0.04 [−0.03; 0.13] |
| Africa | 0.16; 0.05 [0.07; 0.26] | 0.09; 0.04 [0.01; 0.18] |
| Latinos | 0.14; 0.02 [0.05; 0.24] | 0.14;0.02 [−0.06; 0.11] |
| Middle East | 0.10; 0.05 [0.10; 0.31] | 0.05; 0.04 [−0.01; 0.14] |
| West European/American | 0.08; 0.06 [−0.03; 0.18] | 0.01; 0.05 [−0.07; 0.12] |

Legend: *CI* = confidence interval; *LL* = lower limit; *UL* = upper limit.

## 3. Discussion

The study's primary objective was to examine a more complex model by linking the binding moral foundations, perception of realistic and symbolic threats, explicit prejudice against migrants, and indexes of social distance from several migrant groups.

In line with previous studies (e.g., Baldner and Pierro 2019; Kugler et al. 2014), we predicted that individuals who endorsed the binding moral foundations would be more prejudiced against migrants, and that this relation could be mediated by perceived realistic or symbolic threats from migrants. Building on previous research (Hadarics and Kende 2017), we also expected that individualizing moral foundations would be negatively correlated with prejudice against migrants. We found support for both hypotheses in both countries using the index of explicit prejudice. However, when social distance was used as an index of prejudice, more complex results emerged. These showed that the relationship between the binding moral foundations and social distance/prejudice was not always linear and significant. More specifically, there was no direct relation between the binding moral foundations and social distance in five out of the six migrant groups in both countries, but the relationship was mediated through perceived realistic threat for all the migrant categories, except for the West European/American migrants. The indirect effect of the binding moral foundations on social distance, through perceived realistic threat, was stronger for African and Middle Eastern migrants in Malta, and for Asian and African migrants in Italy. The results confirmed that the participants had a large social distance from these groups. Therefore, our results support the assumption that the binding moral foundations work as the bases for negative attitudes toward migrants (e.g., Hadarics and Kende 2017), and that the effect of the binding moral foundations on prejudiced attitudes increases, especially when a threat to the ingroup's wealth/resources/security is perceived (i.e., perceived realistic threat). More specifically, individuals who are loyal to their community, respect the norms of authority, and see purity as relevant (i.e., binding moral foundations) are more likely to perceive migrants as a realistic threat, and to distance themselves from them, but not equally from all groups. The effect of the binding moral foundations on social distance from the West European/American migrants is not mediated by perceived threat. The social distance from this group of migrants is lower, and they are less perceived as a threat.

The effects of the binding moral foundations on social distance indexes were not mediated through perceived symbolic threat in the Maltese sample, whilst in the Italian sample they were mediated only for the African migrants. These results imply that,

when mediated by perceived realistic threat, the relationship between the binding moral foundations and the index of social distance is significant for the majority of migrant groups considered in this study, but this is not the case when perceived symbolic threat is considered as a mediator. The effect of perceived symbolic threat is diversified in our samples in relation to different migrant groups.

*Limitations and Further Research*

This study has several limitations. The majority of the respondents in both samples were female (Malta: 70.2%; Italy: 57.6%), and young (41% were 18 to 30 years old). Inclusion criteria were being born Maltese and Italian, and being aged 18 years and over. In order to make the questionnaire as anonymous as possible, we did not ask participants to state their age, but instead to indicate their age group, as age, together with some socio-demographic characteristics, could have been considered by the respondents to be an indirect identifier. Maltese and Italian residents living abroad were excluded.

Since Italian students participating in the study were asked to distribute five questionnaires and were explicitly instructed to distribute these amongst acquaintances, we are confident they did so, but we had no control over whether they had effectively distributed the questionnaire to third parties. However, on analyzing our data, we observed normal distributions. This increases our confidence that the questionnaires were disseminated as requested.

Further research could investigate the relationship between the binding and individualizing moral foundations, perceived threat, and prejudice against specific groups of migrants. Our study measured attitudes toward some macro-categories of migrants of different regional provenance, but we are aware that there could be different attitudes toward more specific ethnic groups within each macro-category. Moreover, a recent study has gone even further, showing that there is heterogeneity in attitudes toward a single migrant group (Segersven et al. 2023). Attitudes that the majority hold toward migrants are not "one size fits all": one group of migrants may be closer to the majority in some aspects (e.g., use of State-provided services); however, the same migrant group might not have any contact with the majority in some other aspects (e.g., family and traditional customs). Thus, future studies should take into consideration these additional aspects by using several measures of prejudice, including the implicit association test. Furthermore, the results regarding low partial mediation suggest that other mediators could be at work. Further studies could also investigate the change in prejudice over a prolonged period (longitudinal studies) to establish whether changes in migration figures contribute to a change in prejudice against migrants. Experimental studies that manipulate threat perception could corroborate these results, adding a valid contribution to what is already established in the literature.

## 4. Materials and Methods

### 4.1. Participants

Three hundred and eighty respondents participated in the survey. Two different samples were gathered, from Malta ($N$ = 191) and Italy ($N$ = 189), in the period from February to May 2020. In the Maltese sample, 70.2% of the participants were female, and in the Italian sample, 57.6% were female. The inclusion criteria were being born Maltese or Italian, and being aged 18 years and over. The majority of respondents were 18 to 30 years old (41%). Maltese and Italian residents living abroad were excluded. We conducted a power analysis (Gpower 3; Faul et al. 2009) considering 0.05 as a threshold probability to reject the null hypothesis and the expected correlation (r = 0.20, or the equivalent incremental $f^2$ within multiple regression $f^2$ = 0.0416); in this study, 95% power was achieved with $n$ = 266; therefore, acceptable levels of power were met with our sample.

### 4.2. Procedure

Participants were contacted through a message posted on the researcher's social network platforms (Facebook, LinkedIn). A questionnaire using the instruments described

below was presented in a fixed sequence. On submission, responses were registered anonymously. Some of the Italian participants were students attending a course in social psychology. They were asked to distribute five questionnaires and, in return, were given a course credit. Participation was voluntary. Informed consent was sought and obtained from all participants. The research project was approved by the Ethical Committee in the department (BLINDED for review).

*4.3. Measures*

A questionnaire was used containing the following scales and items relative to some socio-demographic characteristics (age, gender, level of education).

We used the Moral Foundations Questionnaire (MFQ; Graham et al. 2008). The original questionnaire in English was administered to Maltese participants, and the Italian version was administered to Italian participants (Bobbio et al. 2011). This scale is available at www.moralfoundations.org (accessed on 1 March 2023). The first section of the MFQ contains 15 items measuring moral relevance, while the second section contains another 15 items that assess moral judgments (3 items for each moral foundation: i.e., care, fairness, loyalty, authority, purity). In the first section, the participants were asked to indicate on a 6-point Likert scale from 1 (not at all relevant) to 6 (extremely relevant) to what extent they considered the proposed items relevant when deciding whether something was right or wrong. Examples for the moral foundations include the following: whether or not someone has suffered emotionally; whether or not someone has acted unfairly; whether or not someone has shown a lack of loyalty; whether or not someone has shown a lack of respect for authority; and whether or not someone has done something disgusting. In the second section, the participants were asked to indicate their level of agreement on a scale ranging from 1 (strongly disagree) to 6 (strongly agree) with statements supporting or rejecting foundation-related judgments. Examples for the moral foundations include the following statements: Compassion for those who are suffering is the most crucial virtue; I think it's morally wrong that rich children inherit a lot of money while poor children inherit nothing; Respect for authority is something all children need to learn; People should be loyal to their family members, even when they have done something wrong; and I would call some acts wrong on the grounds that they are unnatural. A composite index was calculated for each of the five moral foundations by combining the relevance and judgment scores. Finally, two indexes were calculated: one for the binding dimension (comprehending loyalty, authority, and purity foundations) and another for the individualizing dimension (including care and fairness foundations). High scores indicate high levels of moral foundations. Internal reliability was satisfactory as follows: Malta: binding $\alpha = 0.88$; individualizing $\alpha = 0.75$; and Italy: binding $\alpha = 0.87$; individualizing $\alpha = 0.79$.

The Scale of Perceived Threat. A 10-item scale was developed by considering affirmations from various mass media and political discourses that are shared by the two nations. These 10 items reflected realistic and symbolic threats. Examples: Malta/Italy is a country invaded by migrants; Criminality has increased since migrants started coming to Italy/Malta; Migrants have values that are very different to ours; A multiethnic society that so many advocate for shall be fatal and will destroy our culture. Participants were asked to estimate their perception of threat on a range from 1 (completely disagree) to 6 (completely agree). From the Principal Axis Factoring, a bi-factorial structure emerged: Malta 66.78% of the variance (55.60% first component, 10.80% second component), Italy 66.16% of the variance (54.73% first component, 11.44% second component). We created two indexes in each country. The Cronbach alpha was high as follows: Malta: $\alpha = 0.89$ and $\alpha = 0.81$, respectively; and Italy: $\alpha = 0.90$ and $\alpha = 0.81$, respectively.

In terms of the Explicit Prejudice Scale, in order to measure prejudice as a dependent variable, we created an eight-item scale by extrapolating public discourse statements (examples: Migrants and Italians can never feel comfortable with each other; I would not be happy if my children were to befriend migrants; It irritates me when migrants do not speak Italian well) Participants were asked to assess their attitudes on a scale ranging from 1 (completely

disagree) to 6 (completely agree). On examining the factorial structure, a mono-factorial structure emerged and all the communalities were above 0.37. As a result, an index of explicit prejudice was created; high scores indicate high levels of explicit prejudice toward migrants. Internal reliability was high as follows: Malta: $\alpha = 0.86$; and Italy: $\alpha = 0.92$.

Regarding Bogardus' social distance scale (Bogardus 1933), we used this as an additional measure of prejudice as it captures the behavioral components of prejudice (e.g., Doell 2006). We reasoned that the distance individuals choose to put between themselves and migrants reflects their prejudice toward them. Participants were required to indicate the extent to which they would be accepting each of six groups of migrants (Asian, East European, African, Hispanic, Middle Eastern, and West European/American migrants) on a 6-point scale of responses (1 = Would accept as close relatives by marriage (i.e., as the legal spouse of a close relative); 2 = as close personal friends; 3 = as neighbours on the same street; 4 = as co-workers in the same occupation; 5 = only as visitors in my country; 6 = Would exclude from entry into my country). We calculated indexes of social distance for each migrant group by summing the positive responses and subtracting this sum from the negative category "no relations". Thus, high scores indicate high social distance.

## 5. Conclusions

This research has some theoretical and practical implications for both migrant and autochthonous populations. Both countries in this study have high migration rates: Malta has the highest migration rate in Europe (46.3 per 1000 inhabitants), while Italy's migration rate is somewhat lower (5.7 per 1000 inhabitants). Italy's migration rate from outside the European Union (69.9%) is higher than that of Malta (39%) (Eurostat 2019). The results of this study demonstrate that prejudice against migrants is slightly higher in Malta than in Italy, congruent with migration rates.

This study expanded on the finding of previous studies that found a relationship between the binding moral foundations and prejudice toward migrants. Here, we have shown that this relationship is not stable for all migrant groups and that it is mediated by the perception of migrants as a threat. Perceived realistic threat emerged as more significant mediator than perceived symbolic threat. The stronger the binding morality of the autochthonous group, the likelier the perception the immigrants pose a realistic threat for the community, and the liker it is that the people who base their morality on the binding moral foundations will be prejudiced toward migrants. However, this study also highlights that the social distance between the autochthonous group and immigrant groups is not homogenous, as some immigrant groups seem to pose a higher degree of perceived realistic threat and, as such, more social distance is kept from them. This study reiterates the need to analyze in our societies which groups are perceived as the greatest threat and for what reasons, and to reflect together with political leaders what should be done in order to develop inter-group relations based on the values of respect and tolerance.

**Author Contributions:** Conceptualization: A.K.; Methodology: A.K. and F.B.; Software: A.K. and F.B.; Validation: A.K.; Formal analysis: A.K. and F.B.; Investigation, F.B.; Data curation: A.K. and F.B.; Writing—original draft preparation: A.K. and F.B.; Writing—review and editing: A.K.; Supervision, A.K.; Project administration, A.K. All authors have read and agreed to the published version of the manuscript.

**Funding:** This research received no external funding.

**Institutional Review Board Statement:** The study was conducted in accordance with the Declaration of Helsinki, and approved by the Ethics Committee of the Department of Developmental and Social Psychology (Dipartimento di Psicologia dei Processi di Sviluppo e Socializzazione) (protocol code. 0000789 and date of approval 3 June 2019).

**Informed Consent Statement:** Informed consent was obtained from all subjects involved in the study.

**Data Availability Statement:** Dataset will be available upon request to the authors.

**Conflicts of Interest:** The authors declare no conflict of interest.

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
