# Peer review of "The Effects of Binding Moral Foundations on Prejudiced Attitudes toward Migrants: The Mediation Role of Perceived Realistic and Symbolic Threats"

_genealogy, doi:10.3390/genealogy7030065_

Round 1

Reviewer 1 Report

A very important study with obvious ramifications!

Author Response

Dear Reviewer, 

thanks a lot for your kind feedback.

All the best.

Reviewer 2 Report

While the paper does provide a background on the topic, it could benefit from a more in-depth exploration and contextualization of previous research. This would help readers understand the nuances and gaps in the existing literature that the current study aims to address.

The research design, questions, and hypotheses should be presented more explicitly. It would be beneficial to have a clear and concise section dedicated to outlining the research questions and hypotheses, ensuring that readers can quickly grasp the study's primary objectives and predictions.

The discussion of findings and conclusions could be more compelling. The authors should ensure that their arguments are coherent and that they draw clear connections between their results and the broader implications for the field. Additionally, they should emphasize how their findings contribute to or challenge existing knowledge on the topic.

While the English language quality in the paper is generally good, there are areas where clarity can be improved. An in-depth language review might help refine the paper, ensuring that complex ideas are communicated effectively.

Author Response

Dear Reviewer,

Thanks for your attention, time, and constructive comments for improving our manuscript, as well as for the opportunity to revise the manuscript and submit a new version of it.

In revising the manuscript, we considered all the comments that you raised. Please, note that in this revised version of the manuscript, we used track changes so that it will be convenient for you to see the changes from our prior submission.

We hope that the revised version meets all your requests and criteria.

Your comments:

While the paper does provide a background on the topic, it could benefit from a more in-depth exploration and contextualization of previous research. This would help readers understand the nuances and gaps in the existing literature that the current study aims to address.

The research design, questions, and hypotheses should be presented more explicitly. It would be beneficial to have a clear and concise section dedicated to outlining the research questions and hypotheses, ensuring that readers can quickly grasp the study's primary objectives and predictions.

We added a paragraph at the end of introduction presenting the research questions more explicitly.

The discussion of findings and conclusions could be more compelling. The authors should ensure that their arguments are coherent and that they draw clear connections between their results and the broader implications for the field. Additionally, they should emphasize how their findings contribute to or challenge existing knowledge on the topic.

We reviewed the discussion and conclusions.

Comments on the Quality of English Language

While the English language quality in the paper is generally good, there are areas where clarity can be improved. An in-depth language review might help refine the paper, ensuring that complex ideas are communicated effectively.

We have done language review, but if it is still not satisfying, we will ask for more time in order to be able to send the manuscript to a proofreading service.

Reviewer 3 Report

Dear author(s),

Thank you for the interesting paper! The study explores attitudes toward migrants in Malta and Italy. The main contribution of the manuscript is seeing attitudes on migration not as a single block but several blocks (6 in this paper). A recent study has gone even further to show that there is a heterogeneity in attitudes toward a single migrant group (Segerven et al. 2023). Attitudes that the majority have made out of migrants are not “one size fits for all”: one group of migrants may be closer to the majority in some respect (e.g., both receive public school education); the same migrant group might not have any contact with the majority in some other respect (e.g., family and local customs). Perhaps the author(s) can add this fact to the Introduction and/or 3.1. (lines 273-274) 

We know that the respondents whose attitudes are being investigated were mostly female, young and born in Malta/Italy. But the paper lacks proper descriptive statistics on participants! Also ‘education’ is a relevant potential confounding variable that is not taken into account. Please add some lines to section 3.1. Limitations of the study. 

The manuscript should follow the IMRD structure: Materials and Methods precede Results and Discussion.

Other comments

  • line 27: What do you mean by “binding morality”? Concepts should be defined before using them.

  • line 67: “that” => “than”

  • line 122: “binging” => “binding”

  • line 124: “one index of prejudice” add: “in Malta”

  • line 128: Table 1. What is the scale?

  • line 151: “ad exception”

  • line 222: Table 5. Legend is missing.

  • line 281: When was the survey open (i.e. when was the data collected)?

  • line 285: Faul et al. (2009) is missing from References

  • lines 298-326: Please add some examples of items.

  • lines 333-338 should be revised or deleted as the paper has nothing to do with public speakers, political leaders or journalists. Rather, the abstract mentions theoretical and practical implications - how about writing those?

Reference

Segersven, Otto Erik Alexander, Ilkka Ari Tapani Arminen & Mika Simonen (2023) Acculturation Among Finnish Somalis: An Imitation Game inquiry into bicultural fluency. International Migration Review. First published online March 7, 2023. https://doi.org/10.1177/01979183231154555

See above.

Author Response

Dear Reviewer,

Thanks for your attention, time, and constructive comments for improving our manuscript, as well as for the opportunity to revise the manuscript and submit a new version of it.

In revising the manuscript, we considered all the comments that you raised. Please, note that in this revised version of the manuscript, we used track changes so that it will be convenient for you to see the changes from our prior submission.

We hope that the revised version meets all your requests and criteria.

Your comments:

Thank you for the interesting paper! The study explores attitudes toward migrants in Malta and Italy. The main contribution of the manuscript is seeing attitudes on migration not as a single block but several blocks (6 in this paper). A recent study has gone even further to show that there is a heterogeneity in attitudes toward a single migrant group (Segerven et al. 2023). Attitudes that the majority have made out of migrants are not “one size fits for all”: one group of migrants may be closer to the majority in some respect (e.g., both receive public school education); the same migrant group might not have any contact with the majority in some other respect (e.g., family and local customs). Perhaps the author(s) can add this fact to the Introduction and/or 3.1. (lines 273-274) 

Thanks for having suggested this interesting paper. We added this paragraph in Discusion (line 274).

We know that the respondents whose attitudes are being investigated were mostly female, young and born in Malta/Italy. But the paper lacks proper descriptive statistics on participants! Also ‘education’ is a relevant potential confounding variable that is not taken into account. Please add some lines to section 3.1. Limitations of the study. 

As explained in lines 166 – 168, initially we have controlled for individualizing moral foundations, age, and education level. However, we found that age and education have little or no effect on the mediators and dependent variables in both samples. Consequentially, we ran the model using only the index of the individualizing moral foundations as a covariate.

The manuscript should follow the IMRD structure: Materials and Methods precede Results and Discussion.

We followed the structure as suggested in the MDPI template.

Other comments

  • line 27: What do you mean by “binding morality”? Concepts should be defined before using them.

We changed that term with the “binding moral foundations” which has been defined in the introduction (line 50-51).

  • line 67: “that” => “than”

Corrected.

  • line 122: “binging” => “binding”

Corrected.

  • line 124: “one index of prejudice” add: “in Malta”

Done.

  • line 128: Table 1. What is the scale?

We specified in the title.

  • line 151: “ad exception”

Corrected.

  • line 222: Table 5. Legend is missing.

We added the legend.

  • line 281: When was the survey open (i.e. when was the data collected)?

We specified the period od data collection.

  • line 285: Faul et al. (2009) is missing from References

Added.

  • lines 298-326: Please add some examples of items.

Done.

  • lines 333-338 should be revised or deleted as the paper has nothing to do with public speakers, political leaders or journalists. Rather, the abstract mentions theoretical and practical implications - how about writing those?

We deleted this portion, and reviewed the discussion and conclusions.